# Deep Discriminative Structure Proxy Hashing for Cross-modal Retrieval

Kun Cheng [1]   Qibing Qin [2]   Lei Huang [3]

## Abstract

Existing proxy-based hashing methods optimize samples toward independently learned proxies using isolated similarity constraints. Although efficient, this design overlooks the fact that proxies are learned jointly but lack explicit relational or competitive interactions during optimization. Consequently, proxy responses to a sample are often accumulated rather than contrasted, leading to weakly defined decision regions and limited discriminative structure in the Hamming space. In contrast, our method organizes multiple proxies into sample-specific relational structures, enabling proxies to interact and compete when responding to each sample. Through structure-guided learning, these interactions explicitly contrast positive and negative proxy responses, thereby shaping clearer and more discriminative decision boundaries. Extensive experiments on standard cross-modal benchmarks demonstrate that this structured discrimination consistently improves retrieval accuracy and embedding separability. The source code is available at https://github.com/QinLab-WFU/DDSPH.

## 1. Introduction

Cross-modal retrieval aims to efficiently search semantically related data across different modalities, such as retrieving images given textual queries or vice versa (Liu et al., 2019; Hu et al., 2023; Pu et al., 2025b). With the rapid growth of multimodal data in real-world applications, hashing-based cross-modal retrieval has attracted extensive attention due to its low storage cost and fast retrieval speed (Cheng et al., 2025; Li et al., 2025a; Pu et al., 2025a). By learning compact binary codes in a shared Hamming space, cross-modal

[1]School of Computer Science, Qufu Normal University, Rizhao, China [2]School of Computer Engineering, Weifang University, Weifang, China [3]Faculty of Information Science and Engineering, Ocean University of China, Qingdao, China. Correspondence to: Qibing Qin <qinbing@wfu.edu.cn>.

*Proceedings of the 43rd International Conference on Machine Learning*, Seoul, South Korea. PMLR 306, 2026. Copyright 2026 by the author(s).

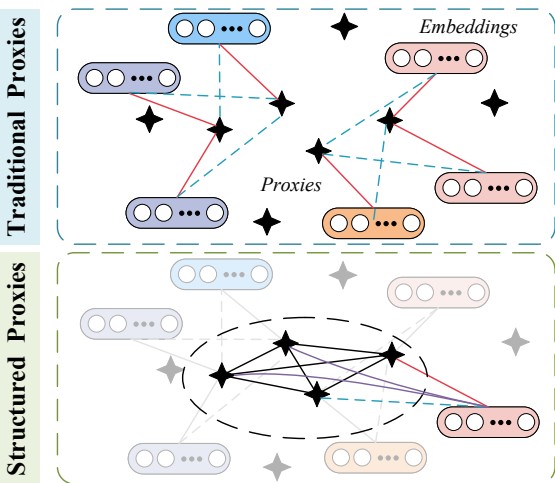

*Figure 1.* Motivation of the proposed structure-discriminative proxy hashing. Conventional proxy-based methods treat proxies as independent semantic anchors, where multiple proxies may simultaneously attract a sample without explicit competition, resulting in overlapping decision regions. By organizing sample-specific proxies into structured and competitive relations, the proposed method explicitly contrasts positive and negative proxy responses, leading to clearer and more discriminative decision boundaries.

hashing methods enable scalable retrieval while preserving semantic consistency across heterogeneous modalities (Luo et al., 2023; Shen et al., 2015; Sun et al., 2023).

In recent years, proxy-based hashing has emerged as a dominant paradigm (Bai et al., 2024; Tu et al., 2023). By introducing learnable category proxies, these methods replace sample-level pairwise or triplet supervision with proxy-level interactions, significantly improving training efficiency and scalability (Huo et al., 2023). Building upon this paradigm, several representative works further incorporate semantic structures into proxy learning (Huo et al., 2024a; Tu et al., 2023; Huo et al., 2024b). These approaches have demonstrated that semantic structures play an important role in organizing representations. Despite these advances, existing proxy-based methods primarily focus on preserving semantic relationships, while the role of such structures in shaping discriminative decision boundaries remains insufficiently explored. In multi-label retrieval, samples often share partially overlapping semantics and are simultaneously associated with multiple labels (Zou et al., 2021). Under this setting,

preserving neighborhood consistency or hierarchical organization does not necessarily guarantee clear discriminative boundaries (Liu et al., 2024; Han et al., 2025). As a result, proxies may act as independent semantic attractors, leading to ambiguous decision regions where samples are difficult to separate, ultimately degrading retrieval reliability.

As shown in Fig. 1, traditional proxy-based methods pull multi-label samples toward multiple independent proxies, forming overlapping semantic regions with weakly defined boundaries. Although semantic relations are preserved, the learned embedding space lacks sufficient discriminative structure. In contrast, structured proxies depicts our key observation that semantic structures should not only be preserved but should actively participate in shaping proxy discrimination. By injecting structural cues into the proxy learning process, discriminative boundaries can be explicitly reinforced, enabling more reliable separation of overlapping semantics across modalities.

Motivated by this observation, we argue that the limitation of existing proxy-based hashing lies not in how semantic relations are defined, but in how proxies interact during optimization. Without explicit interaction or competition, proxy responses to a sample are accumulated rather than contrasted, resulting in weakly discriminative decision regions. To address this issue, we propose Deep Discriminative Structure Proxy Hashing (DDSPH), a structure-discriminative proxy hashing framework for cross-modal retrieval. Specifically, for each training sample, DDSPH selects multiple semantically relevant proxies and organizes them into a sample-specific proxy relations, where nodes correspond to proxies and edges encode their relational roles with respect to the current sample. This proxy relations serves as a structural constraint that governs how proxy responses are compared during optimization, enabling explicit competition between positive and negative proxies. By integrating this structure into proxy learning, DDSPH prevents indiscriminate accumulation of proxy similarities and encourages the formation of clearer, more discriminative decision boundaries in the Hamming space.

The contributions of this work are summarized as follows.

1. Firstly, we propose a novel DDSPH framework, which is the first attempt to the best of our knowledge, to introduce sample-specific proxy structuring into hashing and enables discriminative similarity retrieval across image and text modalities.

2. Secondly, we design discriminative proxy structures toward individual samples by organizing semantically relevant proxies into relational structures with explicit competition, which effectively suppresses proxy response accumulation and sharpens decision boundaries under multi supervision.

3. Extensive experiments on multiple benchmarks demonstrate that the proposed method consistently outperforms SOTA approaches and achieves strong discrimination and retrieval performance without relying on a large number of proxies.

## 2. Related Works

### 2.1. Discriminative Cross-Modal Hashing

Early deep cross-modal hashing methods focus on learning unified binary representations by preserving semantic similarity across heterogeneous modalities, such as Deep Cross-Modal Hashing (DCMH) align image and text representations in a shared Hamming space through pairwise or adversarial supervision, enabling efficient large-scale retrieval (Jiang & Li, 2017; Li et al., 2018; Sun et al., 2024). To further enhance discriminative capability, subsequent studies introduce stronger supervision signals and architectural designs. Differentiable Cross-modal Hashing via Multi-modal Transformers (DCHMT) leverages cross-modal attention to model fine-grained semantic interactions before discretization (Tu et al., 2022), while Modality-Invariant Asymmetric Networks (MIAN) learn asymmetric modality-specific mappings to improve class separability (Zhang et al., 2022).

Beyond direct sample alignment, several works incorporate semantic structure to organize representations(Qin et al., 2024). Related efforts such as Deep Evidential Hashing (DECH) further enhance reliability by modeling uncertainty in semantic supervision (Li et al., 2025b). Diverse Instances Matching for Cross-modal Hashing (DIMCH) explicitly models diverse instance correspondences (Tu et al., 2025). Despite their effectiveness, these approaches enforce discrimination primarily through sample–sample interactions or semantic organization, without explicitly regulating how multiple semantic representatives respond to a given sample.

### 2.2. Proxy-based Learning

Proxy-based learning has emerged as an efficient alternative to pairwise or triplet supervision by introducing learnable semantic proxies to represent class-level information. ProxyNCA and ProxyAnchor replace exhaustive sample comparisons with proxy–sample alignment, significantly improving scalability while maintaining competitive discriminative performance (Movshovitz-Attias et al., 2017; Kim et al., 2020). Beyond independent proxy supervision, ProxyGML explores relational constraints among multiple proxies to propagate semantic information (Zhu et al., 2020). SoftTriple further assigning multiple proxies to each class to capture intra-class variability (Qian et al., 2019). Proxy-based learning has also been adapted to hashing (Tu et al., 2023; Bai et al., 2024). Deep Semantic-aware Proxy Hashing (DSPH) aligns samples with semantic proxies to

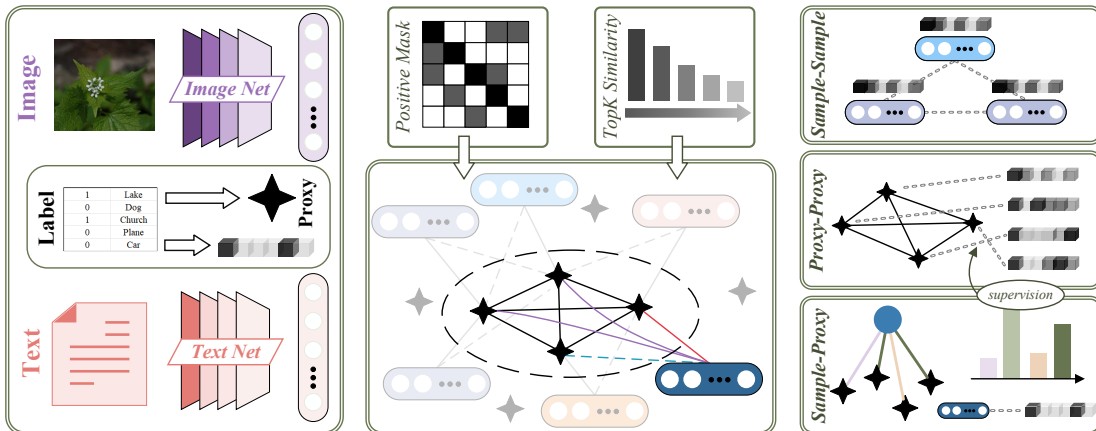

*Figure 2.* Overview of the proposed DDSPH framework. For each sample, the structure-guided relations is constructed by selecting a subset of semantically relevant proxies. Proxy responses are aggregated to produce label-aware predictions, enabling structured discrimination among proxies and joint optimization of hash representations and proxy structures.

achieve efficient cross-modal retrieval (Huo et al., 2023), while Deep Neighborhood-aware Proxy Hashing (DNPH) integrates neighborhood constraints to preserve local semantic consistency (Huo et al., 2024a). Deep Hierarchy-aware Proxy Hashing (DHaPH), further organize proxies according to latent semantic structures to enhance robustness under complex semantics (Huo et al., 2024b). Despite their efficiency, existing proxy-based methods generally treat proxies as independent semantic attractors. During optimization, similarities between a sample and multiple proxies are accumulated rather than explicitly contrasted, which may lead to ambiguous decision regions when samples are associated with overlapping or partially shared semantics.

## 3. Methodology

### 3.1. Problem Definition

In this work, we focus on similarity retrieval across image and text modalities. Let $\mathcal{D} = \{x_i^I, x_i^T, y_i\}_{i=1}^N$ denote a $N$-pairs cross-modal dataset, where $x_i^I$ and $x_i^T$ are paired image and text samples, and $y_i \in \{0, 1\}^C$ is the corresponding multi-hot label vector over $C$ categories from dataset. The goal of cross-modal hashing is to learn modality-specific hashing functions $\mathcal{H}^I, \mathcal{H}^T$ that map heterogeneous inputs into a shared Hamming space $\mathbb{H}$, such that semantically relevant samples across modalities can be efficiently retrieved via $K$-bit binary codes $B^I, B^T$.

Following proxy learning paradigms, we introduce a set of learnable proxies $\mathcal{P} = \{p_j\}_{j=1}^{C \times N}$, where each category is associated with $N$ proxies to capture intra-class diversity. Unlike conventional proxy hashing methods that treat proxies as independent semantic anchors, our objective is to learn a structure-discriminative embedding space and proxies are organized by their semantic relations.

Since discrete optimization is an NP-hard problem, we follow the scheme of previous deep hashing methods by using the $tanh$ function to learn binary-like codes during training:

$$\tilde{B}_i^* = \tanh\left(\mathcal{H}^*(x_i^*; \mathcal{P})\right) \in (-1, 1)^K, * \in \{\mathcal{I}, \mathcal{T}\}. \quad (1)$$

After training, we use $sign$ function to turn it into real binary code $B^I, B^T$ by:

$$B_i^* = \text{sign}(\tilde{B}_i^*) \in \{-1, 1\}^K, * \in \{\mathcal{I}, \mathcal{T}\}. \quad (2)$$

Our framework is shown in Fig. 2.

### 3.2. Feature Extraction

In hash learning, we aim to acquire binary codes that are also low-dimensional to meet efficiency requirements. Therefore, we further compress the extracted features to a specific dimension, such as 16 or 32 bits. Consistent with current mainstream approaches, we employ CLIP, which consists of two transformers $T^\mathcal{I}, T^\mathcal{T}$ as the backbone and add hash layers $H^\mathcal{I}, H^\mathcal{T}$ composed of linear projection layers to further compress the output features to $K$-dimension. During training, the final layer of the hash layer employs a $tanh$ function, with an additional $sign$ layer added during retrieval:

$$B_i^* = \mathcal{H}^*(x_i^*) = H^*\left(T^*(x_i^*)\right), * \in \{\mathcal{I}, \mathcal{T}\}. \quad (3)$$

Since we aim for the hashing layer to produce a final modality-agnostic output, the hash code itself serves as the direct optimization objective.

### 3.3. Relation Construction

Considering that graphs excel at modeling global data affinity, we propose to characterize overall neighbor relationships among samples and proxies with graphs to enhance

the discrimination of proxies. We construct a directed graph $\mathcal{G}$ to contain them, where each node denotes a sample like $x_i^{\mathcal{I}}, x_i^{\mathcal{T}}$ or a proxy like $p_j$, and each edge represents the similarity between two connected nodes. We measure the similarity between samples and proxies by cosine similarity:

$$s_{ij} = \frac{B_i^\top p_j}{\|B_i\|\|p_j\|}. \qquad (4)$$

In this manner, we generate a mini-batch similarity matrix $S \in \mathbb{R}^{B \times CN}$ where $B$ denotes the size of mini-batch, and each row represents the similarity relationship between each sample and all proxies. We can further construct local relationship subgraphs around each sample to enhance granularity. To achieve this, we retain the $k$ maximum values in each row of $S$ to construct the k-nearest neighbor subgraph. Since the number of proxies $N$ may vary, we use a ratio $r \in (0, 1]$ to determine the specific value of $k(= C \times N \times r)$. Since all proxies are initialized randomly, we simultaneously introduce positive masks from their labels to ensure that the selected neighbors are all positive proxies, thereby guaranteeing the subsequent optimization objective. Guided by these two constraints, we transform each row of $S$ into a sparse adjacency matrix $W \in \mathbb{R}^{B \times CN}$, thereby constructing an adjacency graph linking each sample to its nearest positive proxies.

### 3.4. Label Aggregation

After constructing the relation-guided subgraphs, we leverage proxy semantics to produce label-aware predictions for each sample through aggregation. Unlike conventional classification schemes that directly map samples to category scores, we formulate prediction as a proxy-mediated aggregation process. Existing proxy-based methods typically compute sample–proxy similarities independently and directly accumulate their responses when supervising multi-label samples. Under this paradigm, responses from multiple relevant proxies are summed without explicit contrast, which often leads to proxy response accumulation and ambiguous supervision signals. In contrast, our aggregation is performed over a structure-guided subgraph, where proxy responses are selectively weighted and implicitly contrasted through relational constraints. As a result, the aggregated prediction does not merely accumulate proxy activations, but reflects structured competition among related proxies.

Given the subgraph of a sample, we aggregate the semantic information from its connected proxies by weighting their category indicators with the corresponding similarity scores. Let $y_p$ denote the label vector of proxies. The aggregated prediction $z_i$ for a sample $x_i^*$ is computed as

$$z_i = \sum_{j \in W_i} s_{ij}\, y_{p_j}. \qquad (5)$$

The aggregation is normalized by the subgraph degree to ensure scale invariance.

This formulation enables label information to influence sample predictions through the learned proxy relations, effectively performing label aggregation over the constructed subgraph. Importantly, labels are not explicitly propagated as node features but instead act as supervisory signals that shape the proxy embeddings and their associations with samples. This design allows the aggregation process to remain fully differentiable while avoiding explicit label diffusion, making it well-suited for end-to-end optimization in deep hashing frameworks.

### 3.5. Hash Learning

Based on the constructed sample–proxy relations and the aggregated semantic predictions, we formulate the hash learning objective by jointly optimizing sample representations and structured proxies. The learning process is driven by three complementary objectives, i.e., three pairs of relationships, which respectively align samples with proxies, regularize the internal structure of proxies, and preserve similarity relations between samples. The Alignment loss enforces semantic consistency between samples and their associated proxies under the multi-label setting. Instead of assigning each sample to a single proxy or class, we interpret the aggregated response $z_i$ as a class-wise confidence vector induced by the selected proxies. Specifically, for the $c$-th category, the prediction $z_{i,c}$ is expected to be high if $y_{i,c} = 1$ and low otherwise. The Alignment loss is defined as

$$\mathcal{L}_a = \frac{1}{N} \sum_{i=1}^{N} \sum_{c=1}^{C} \Big( -y_{i,c} \log \sigma(z_{i,c}) - \\ (1 - y_{i,c}) \log \big(1 - \sigma(z_{i,c})\big) \Big), \qquad (6)$$

where $\sigma(\cdot)$ denotes the *sigmoid* function. This objective drives the hash representations to align with semantically relevant proxies while allowing multiple categories to be activated simultaneously. Unlike conventional proxy-based objectives that supervise isolated proxy responses, the proposed alignment loss operates on relation-guided aggregated responses, where proxy interactions are explicitly considered before supervision. This structured aggregation implicitly contrasts competing proxies, preventing uniform activation of all relevant proxies and promoting discriminative dominance.

While the Alignment loss constrains sample–proxy relations, it does not explicitly regulate the internal structure of the proxy set. To encourage meaningful organization among proxies, we introduce a Regularization loss that operates directly at the proxy level. Let $P \in \mathbb{R}^{K \times CM}$ be the matrix of normalized proxies, and define the proxy similarity matrix

*Table 1.* mAP I2T and T2I results (%) of DDSPH and baseline methods on three benchmark datasets w.r.t. four hash bits.

| Task | Method | Reference | MIRFLICKR-25K | | | | NUS-WIDE | | | | MS COCO | | | |
|---|---|---|---|---|---|---|---|---|---|---|---|---|---|---|
| | | | *16* | *32* | *64* | *128* | *16* | *32* | *64* | *128* | *16* | *32* | *64* | *128* |
| | DSPH | TCSVT'23 | 81.29 | 84.82 | 85.41 | 86.58 | 68.30 | 69.79 | 71.62 | 72.09 | 70.44 | 75.10 | 76.94 | 77.81 |
| | TwDH | TMM'24 | 79.71 | 81.47 | 83.19 | 84.37 | 66.83 | 69.34 | 69.95 | 71.94 | 64.29 | 70.04 | 73.08 | 75.44 |
| | DNPH | TOMM'24 | 81.08 | 82.69 | 82.89 | 83.70 | 66.89 | 68.11 | 69.39 | 70.93 | 64.38 | 69.10 | 72.94 | 72.51 |
| Image | DHaPH | TKDE'24 | 82.99 | 84.37 | 85.31 | 85.49 | 69.58 | 70.35 | 71.36 | 71.55 | 72.84 | 74.15 | 74.75 | 75.43 |
| | BiLGSEH | TCSVT'25 | 79.29 | 81.16 | 81.94 | 82.07 | **70.50** | 71.42 | 72.18 | 72.13 | 66.68 | 73.33 | 75.96 | 74.85 |
| ↓ | DECH | AAAI'25 | 79.61 | 83.96 | 83.83 | 84.43 | 66.13 | 71.61 | 71.55 | 72.41 | 63.73 | 64.35 | 66.44 | 68.49 |
| | DPBE | MM'25 | 80.82 | 83.27 | 85.12 | 85.90 | 62.46 | 64.51 | 68.35 | 71.14 | 63.25 | 64.77 | 69.26 | 72.61 |
| Text | DDBH | TCSVT'25 | 84.50 | 85.34 | 86.10 | 86.50 | 69.34 | 71.45 | 72.29 | 72.29 | 71.65 | 74.54 | 76.81 | 78.24 |
| | DDWSH | TMM'26 | 84.05 | 85.96 | 86.48 | 87.00 | 69.53 | 70.85 | 71.80 | 72.46 | **72.08** | **75.15** | 77.32 | 78.59 |
| | **DDSPH** | **OURS** | **84.86** | **85.99** | **86.83** | **87.53** | 70.25 | **71.62** | **72.87** | **73.54** | 69.57 | 75.23 | **77.40** | **78.97** |
| | DSPH | TCSVT'23 | 80.00 | 82.38 | 82.94 | 83.87 | 69.97 | 71.53 | 73.04 | 72.64 | 70.40 | **75.56** | 77.13 | 78.52 |
| | TwDH | TMM'24 | 77.80 | 80.01 | 81.96 | 82.96 | 67.06 | 71.02 | 71.37 | 72.60 | 65.68 | 70.92 | 74.45 | 76.11 |
| | DNPH | TOMM'24 | 80.15 | 81.76 | 81.66 | 82.32 | 68.71 | 69.94 | 71.82 | 71.91 | 64.68 | 70.12 | 73.88 | 72.98 |
| Text | DHaPH | TKDE'24 | 81.48 | 81.65 | 82.29 | 82.79 | 68.78 | 70.54 | 69.98 | 70.42 | 69.35 | 70.69 | 71.54 | 71.87 |
| | BiLGSEH | TCSVT'25 | 80.48 | 82.41 | 83.43 | 83.47 | 70.27 | 70.89 | 72.02 | 73.24 | 68.96 | 73.16 | 75.43 | 74.72 |
| ↓ | DECH | AAAI'25 | 78.69 | 81.85 | 82.23 | 83.67 | 68.28 | **73.05** | 73.15 | 73.18 | 62.11 | 65.27 | 66.97 | 69.15 |
| | DPBE | MM'25 | 79.31 | 81.54 | 83.55 | 84.06 | 63.49 | 66.23 | 69.63 | 73.42 | 62.56 | 64.23 | 71.75 | 74.93 |
| Image | DDBH | TCSVT'25 | 82.45 | 83.18 | 83.90 | 84.33 | 70.23 | 72.11 | 73.25 | 73.53 | **71.67** | 73.94 | 75.95 | 77.07 |
| | DDWSH | TMM'26 | 82.10 | 83.61 | 84.03 | 84.56 | 70.49 | 71.77 | 73.21 | 73.87 | 69.72 | 73.28 | 75.82 | 77.29 |
| | **DDSPH** | **OURS** | **83.22** | **84.05** | **84.90** | **85.52** | **71.31** | 72.91 | **73.81** | **74.31** | 69.82 | 74.66 | **77.36** | **78.98** |

The best and second-best performances are highlighted in **boldface** and underlined.

$S_p$ as

$$S_{ij} = p_i^\top p_j, \quad (7)$$

where $S_{ij}$ measures the cosine similarity between the $i$-th and $j$-th proxies.

The Regularization loss enforces high similarity between proxies belonging to the same category while suppressing similarity across different categories, which is formulated as

$$\mathcal{L}_r = \frac{1}{(CN)^2} \sum_{i=1}^{CN} \sum_{j=1}^{CN} \left( S_{ij} - \mathbb{I}[y_i = y_j] \right)^2, \quad (8)$$

where $\mathbb{I}[\cdot]$ is the indicator function with $\mathbb{I}[x] = 1$ if $x$ is true and $\mathbb{I}[x] = 0$ otherwise. This objective encourages proxies of the same category to form compact clusters in the embedding space, while maintaining inter-category separability. As a result, the proxy set serves as a structured semantic scaffold that stabilizes and regularizes the optimization of sample representations. In addition to proxy-guided supervision, preserving instance-level semantic relations remains crucial for effective hashing. To this end, we introduce a metric-based Preservation loss that directly operates on sample representations.

Given a triplet $(i, i^+, i^-)$, where sample $i^+$ shares at least one semantic label with $i$ and sample $i^-$ is semantically dissimilar, we enforce a relative distance constraint in the continuous Hamming relaxation space. The Preservation loss is defined as

$$\mathcal{L}_p = \sum \max \left( 0, \ m + d(h_i, h_{i^+}) - d(h_i, h_{i^-}) \right), \quad (9)$$

where $d(\cdot, \cdot)$ denotes the cosine distance between normalized hash representations and $m$ is a margin hyper-parameter. This loss encourages semantically similar samples to be mapped closer than dissimilar ones, reinforcing discriminative structure at the instance level. The final training objective combines three complementary loss components:

$$\mathcal{L} = \frac{\mathcal{L}_a}{\text{detach}(\mathcal{L}_a)} + \frac{\mathcal{L}_r}{\text{detach}(\mathcal{L}_r)} + \frac{\mathcal{L}_p}{\text{detach}(\mathcal{L}_p)}, \quad (10)$$

where $\mathcal{L}_a$, $\mathcal{L}_r$, and $\mathcal{L}_p$ correspond to sample–proxy alignment, structure-guided proxy discrimination, and instance-level semantic preservation, respectively. Each loss term is normalized by its detached magnitude to avoid introducing additional weighting hyper-parameters and to ensure balanced multi-objective optimization. In practice, this normalization leads to stable training behavior across datasets without requiring dataset-specific tuning. Through joint optimization, the proposed framework learns discriminative hash codes and structured proxies that enable efficient and accurate cross-modal retrieval.

## 4. Experiment

### 4.1. Datasets and Implementation Details

Three publicly available benchmark cross-modal datasets are utilized to evaluate retrieval performance, namely MIRFLICKR-25K (Huiskes & Lew, 2008), contains 25000 pairs of image-text samples, which we preprocessed into 24581 pairs containing 24 categories, MS COCO (Lin et al., 2014), was created by Microsoft and contains 122218 sam-

*Table 2.* FHD (%) of DDSPH and baseline methods w.r.t. four hash bits on three benchmark datasets. We randomly selected 50, 100, 200, and 400 pairs of positive and negative heterogeneous pairs from the database set for the samples in the query set, repeating each scenario 5 times with different seeds to ensure stability.

| FLICKR* | 16 bits | | 32 bits | | 64 bits | | 128 bits | |
|---|---|---|---|---|---|---|---|---|
| | I→T | T→I | I→T | T→I | I→T | T→I | I→T | T→I |
| DHaPH | 90.15 ± 0.14 | 84.38 ± 0.19 | 104.54 ± 0.14 | 88.16 ± 0.07 | 109.22 ± 0.17 | 90.89 ± 0.21 | 108.28 ± 0.18 | 92.46 ± 0.08 |
| BiLGSEH | 69.81 ± 0.11 | 69.05 ± 0.06 | 76.43 ± 0.17 | 75.86 ± 0.06 | 80.42 ± 0.19 | 82.62 ± 0.07 | 81.36 ± 0.18 | 84.46 ± 0.08 |
| DECH | 81.13 ± 0.13 | 67.85 ± 0.06 | 96.38 ± 0.13 | 80.76 ± 0.15 | 95.41 ± 0.16 | 82.94 ± 0.10 | 91.55 ± 0.10 | 87.98 ± 0.13 |
| DPBE | 82.80 ± 0.10 | 67.31 ± 0.10 | 95.14 ± 0.16 | 75.99 ± 0.13 | 102.19 ± 0.18 | 84.55 ± 0.11 | 103.01 ± 0.10 | 86.81 ± 0.15 |
| DDBH | **100.13 ± 0.12** | 84.51 ± 0.07 | 104.65 ± 0.13 | 84.51 ± 0.11 | 109.21 ± 0.26 | 90.89 ± 0.08 | **111.30 ± 0.10** | 92.47 ± 0.06 |
| DDWSH | 94.35 ± 0.17 | 83.56 ± 0.11 | 104.48 ± 0.14 | 89.38 ± 0.13 | 106.57 ± 0.17 | 88.78 ± 0.11 | 106.79 ± 0.15 | 88.67 ± 0.13 |
| **DDSPH** | 99.05 ± 0.10 | **88.56 ± 0.14** | **105.31 ± 0.14** | **91.91 ± 0.12** | **109.30 ± 0.09** | **94.92 ± 0.12** | 108.34 ± 0.15 | **97.61 ± 0.19** |
| **NUS-WIDE** | | | | | | | | |
| DHaPH | 101.03 ± 0.13 | 98.86 ± 0.16 | 90.15 ± 0.14 | 90.05 ± 0.16 | 107.21 ± 0.15 | 104.04 ± 0.14 | 106.15 ± 0.13 | 103.22 ± 0.13 |
| BiLGSEH | 99.77 ± 0.13 | 101.87 ± 0.11 | 100.32 ± 0.21 | 102.42 ± 0.17 | 99.00 ± 0.18 | 103.24 ± 0.14 | 99.00 ± 0.18 | 103.21 ± 0.13 |
| DECH | 93.92 ± 0.14 | 97.85 ± 0.13 | 105.56 ± 0.14 | 108.89 ± 0.11 | 97.69 ± 0.15 | 105.49 ± 0.10 | 104.69 ± 0.13 | 109.24 ± 0.09 |
| DPBE | 86.57 ± 0.14 | 88.48 ± 0.08 | 91.81 ± 0.12 | 95.98 ± 0.13 | 101.23 ± 0.19 | 102.58 ± 0.11 | 105.16 ± 0.14 | 109.67 ± 0.08 |
| DDBH | **110.09 ± 0.17** | **110.02 ± 0.17** | 112.72 ± 0.17 | 111.14 ± 0.15 | **115.86 ± 0.11** | 113.44 ± 0.14 | **116.49 ± 0.15** | 116.31 ± 0.15 |
| DDWSH | 59.20 ± 0.10 | 42.84 ± 0.12 | 105.50 ± 0.14 | 98.44 ± 0.11 | 70.92 ± 0.12 | 69.57 ± 0.12 | 80.65 ± 0.12 | 78.01 ± 0.12 |
| **DDSPH** | 106.75 ± 0.21 | 109.67 ± 0.12 | **113.42 ± 0.16** | **114.35 ± 0.11** | 114.43 ± 0.16 | **117.81 ± 0.10** | 115.71 ± 0.13 | **117.78 ± 0.17** |
| **MS COCO** | | | | | | | | |
| DHaPH | 116.94 ± 0.17 | 105.43 ± 0.07 | 113.73 ± 0.14 | 114.38 ± 0.11 | 117.05 ± 0.10 | 120.65 ± 0.07 | 120.58 ± 0.15 | 119.58 ± 0.09 |
| BiLGSEH | 90.76 ± 0.11 | 90.07 ± 0.12 | 99.46 ± 0.14 | 100.66 ± 0.14 | 105.93 ± 0.08 | 106.60 ± 0.08 | 106.24 ± 0.12 | 106.50 ± 0.09 |
| DECH | 88.55 ± 0.18 | 90.90 ± 0.05 | 94.46 ± 0.12 | 105.41 ± 0.11 | 95.77 ± 0.10 | 108.17 ± 0.10 | 98.15 ± 0.12 | 107.36 ± 0.11 |
| DPBE | 88.17 ± 0.15 | 86.30 ± 0.13 | 95.19 ± 0.16 | 103.84 ± 0.10 | 106.48 ± 0.14 | 107.95 ± 0.12 | 117.72 ± 0.14 | 120.85 ± 0.14 |
| DDBH | 123.63 ± 0.17 | **124.29 ± 0.18** | 131.57 ± 0.15 | 123.37 ± 0.11 | **136.25 ± 0.21** | 124.64 ± 0.15 | **139.10 ± 0.20** | 126.46 ± 0.10 |
| DDWSH | 123.67 ± 0.08 | 97.88 ± 0.07 | 131.81 ± 0.11 | 111.00 ± 0.15 | 135.53 ± 0.12 | 112.52 ± 0.14 | 137.45 ± 0.17 | 118.76 ± 0.12 |
| **DDSPH** | **126.72 ± 0.19** | 111.66 ± 0.13 | **135.72 ± 0.18** | **127.61 ± 0.08** | 130.46 ± 0.17 | **132.66 ± 0.17** | 138.61 ± 0.09 | **131.09 ± 0.08** |

The best and second-best performances are highlighted in **boldface** and underlined. FLICKR* i.e. MIRFLICKR-25K.

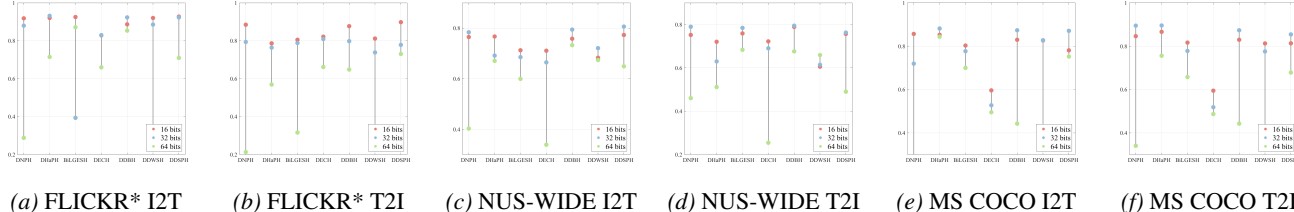

| *(a)* FLICKR* I2T | *(b)* FLICKR* T2I | *(c)* NUS-WIDE I2T | *(d)* NUS-WIDE T2I | *(e)* MS COCO I2T | *(f)* MS COCO T2I |
|---|---|---|---|---|---|

*Figure 3.* $P@H \leq 2$ results of DDSPH and baseline methods on three benchmark datasets. FLICKR* i.e. MIRFLICKR-25K.

ple pairs from 80 categories. And NUS-WIDE (Chua et al., 2009), was constructed by the National University of Singapore, containing 195,834 pairs.

The backbone setup of our proposed DDSPH is based on the open-source PyTorch 2.3.0 framework on NVIDIA RTX A6000 Ada and the pipeline can be referenced as follows (Huo et al., 2024b). All experiments randomly divide the dataset into 10000 pairs of samples as the training set. Then set 5000 pairs as the query set, and the remaining as the database set. In training, the batch size is set to 256, and the training results are taken as the best value within 100 rounds of epoch.

### 4.2. Performance comparison with baselines

We use recent significant cross-modal hashing retrieval works as the basis for evaluating performance, includ-

ing: Deep Semantic-aware Proxy Hashing (DSPH) (Huo et al., 2023), Two-step discrete hashing (TwDH) (Tu et al., 2024), Deep Neighborhood-preserving Hashing (DNpH) (Qin et al., 2024), Deep Hierarchy-aware Proxy Hashing (DHaPH) (Huo et al., 2024b), Bi-Direction Label-Guided Semantic Enhancement Hashing (BiLGSEH) (Zhu et al., 2025), Deep Evidential Hashing (DECH) (Li et al., 2025b), Deep Probabilistic Binary Embedding (DPBE) (Cheng et al., 2025), Deep Discriminative Boundary Hashing (DDBH) (Qin et al., 2025), Deep Distance Weighted Sampling Hashing (DDWSH) (Cheng et al., 2026). All methods use CLIP ViT-B/32 with official OpenAI pretrained weights as the backbone. We compare the proposed DDSPH with SOTA methods on three benchmarks, as shown in Tab. 1. We use mAP@all, the average precision under different recall thresholds, as the primary metric for evaluating detection and retrieval performance. The table demonstrates that

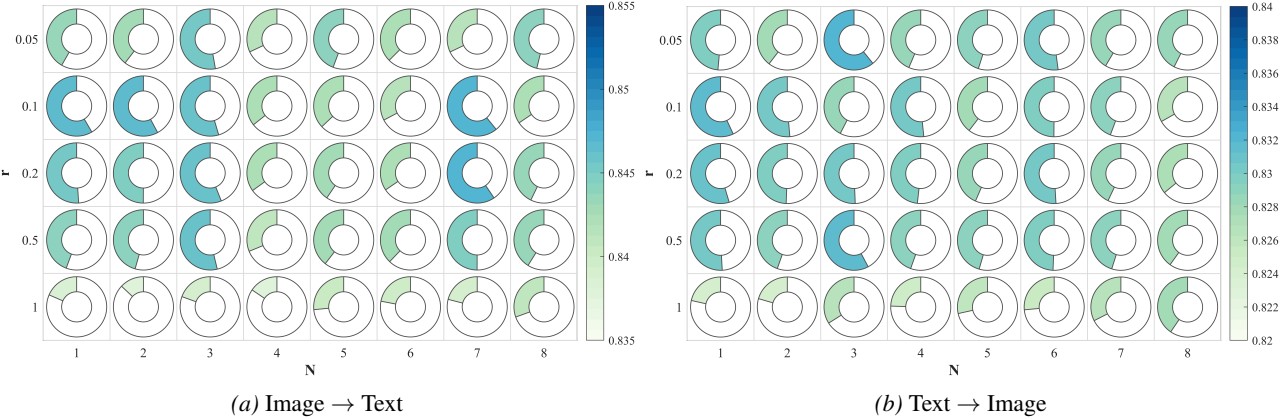

*(a)* Image → Text  *(b)* Text → Image

*Figure 4.* mAP@all I2T and T2I results of hyper-parameter sensitivity analysis on MIRFLICKR-25K 16 bits.

DDSPH achieves the best performance in the majority of settings, with particularly consistent gains on MIRFLICKR-25K and NUS-WIDE across all bit lengths. On MS COCO, DDSPH remains competitive at longer codes while showing a smaller margin at 16 bits, where the higher semantic complexity of 80 categories places greater demand on hash code capacity.

### 4.3. Discriminative Hamming Space

Superior retrieval performance is attributed to the discrimination of the collaborative space. We employ two metrics, the Fisher ratio of P/N Hamming Distance (FHD) and Precision within Hamming Radius $\leq 2$ ($P@H \leq 2$), to assess the discriminative capability of the Hamming space. The former calculates the distance between positive and negative heterogeneous samples in the trained space. The latter is a classic hash learning metric that evaluates the correlation of hash codes within two-bit differences. The results of FHD are shown in Tab. 2. Our method maintains an excellent Fisher ratio, which indicates that positive samples are closer and negative samples are farther in its learned space. Notably, on NUS-WIDE, DDSPH achieves the highest FHD in most bit-length and direction settings, suggesting that the structured proxy competition is particularly effective when semantic categories partially overlap, as is characteristic of this dataset. Furthermore, Fig. 3 shows that our method also achieves preferable results on $P@H \leq 2$, which indicates that negative samples are effectively pushed away. This is consistent with the FHD findings and provides complementary evidence that DDSPH produces well-separated binary codes rather than merely shifting global distance statistics. DDBH also demonstrates outstanding performance on both metrics, as its boundary-aware training objective targets hash discriminability in a similar spirit. The competitive results between DDSPH and DDBH validate that structure-guided proxy competition is an effective alternative mechanism for achieving discriminative Hamming spaces.

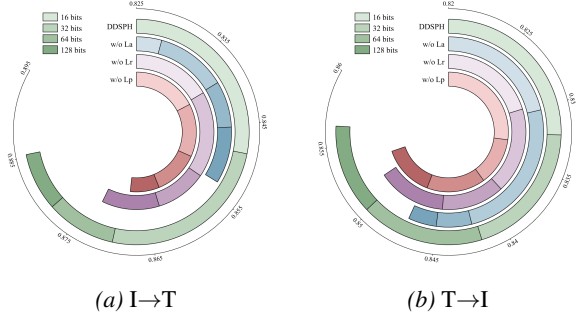

*(a)* I→T  *(b)* T→I

*Figure 5.* mAP@all I2T and T2I results of Ablation study in three loss item w.r.t. four hash bits on MIRFLICKR-25K.

### 4.4. Parameter Analysis and Ablation Study

Traditional proxy methods typically generate only one proxy per class, resulting in a lack of discrimination during training. To address this, we employ multiple proxies to construct a discriminative structure. However, it remains unclear whether this discriminative capability stems from the structure itself or the sheer number of proxies. Therefore, we conduct hyper-parameter experiments by adjusting the hyper-parameters $N$ and $r$ that control the number of proxies, examining the sensitivity of our method to the number of proxies. As shown in Fig. 4, performance remains stable across different proxy combinations, proving that the improvement stems from the discriminative structure rather than simply increasing the number of proxies. As shown, performance remains stable across a wide range of $N$ and $r$ combinations, while $r$=1.0 consistently underperforms sparser settings across all values of $N$. This degradation confirms that structural selectivity—not merely proxy count—is the primary source of improvement: when all proxies are included without filtering ($r$=1), cross-category coupling collapses into undifferentiated accumulation, which is precisely the failure mode DDSPH is designed to address. We set $N$=3 and $r$=0.1 based on a joint consideration of retrieval

performance and training efficiency, as this configuration achieves performance close to larger $N$ settings at roughly two-thirds the training cost.

To further validate the effectiveness of each component in the proposed framework, we conduct ablation studies by removing individual loss terms from the overall objective. Specifically, we evaluate variants without the sample–proxy alignment loss $\mathcal{L}_a$, the structure-guided proxy discrimination loss $\mathcal{L}_r$, and the proxy regularization loss $\mathcal{L}_p$. As shown in Fig. 5, removing any loss term leads to performance degradation, with the most significant drop observed when $\mathcal{L}_a$ is excluded, demonstrating that explicit proxy discrimination is essential for the proposed method. By compared with soft or no masking strategies, the proposed hard positive mask consistently achieves better performance, indicating that filtering irrelevant proxies is crucial for preventing noisy proxy responses and preserving discriminative structure.

To justify the dynamic normalization design in Eq. (10), we compare it against fixed-weight combinations on MIRFLICKR-25K and MS COCO w.r.t. 32 bits. As shown in Table 3, some configuration yields moderate gains for certain tasks, but no fixed configuration matches the dynamic normalization, which automatically brings all three terms to a comparable scale throughout training without dataset-specific tuning.

### 4.5. Noisy Robustness

We further evaluate the robustness of DDSPH under noisy supervision. Following (Wang et al., 2024), we randomly select a fixed proportion of training samples on MIRFLICKR-25K and corrupt their labels by flipping two bits across four selected dimensions, as summarized in Tab. 4. The results show that DDSPH consistently maintains stable retrieval performance under increasing noise levels, especially in high-bit scenarios. This demonstrates that the proposed structure-discriminative proxy organization effectively suppresses the influence of noisy labels by enforcing explicit competition among proxies, leading to more stable decision boundaries and robust semantic alignment.

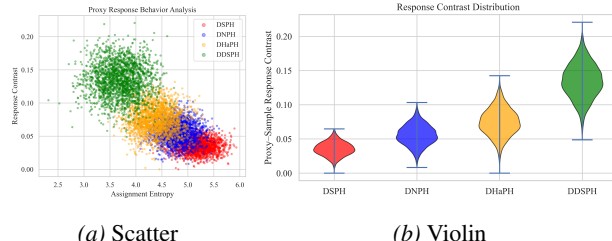

*(a)* Scatter          *(b)* Violin

*Figure 6.* Proxy response analysis on MS-COCO 32bits. (a) Joint visualization of proxy–sample assignment entropy and response contrast. Existing methods show diffuse responses with low contrast, while DDSPH produces more contrasted proxy responses with moderate entropy. (b) Distribution of proxy–sample response contrast, where DDSPH consistently achieves higher contrast across samples.

### 4.6. Proxy Response Analysis

To analyze how proxy-based methods organize the embedding space, we examine proxy–sample response contrast and proxy assignment entropy, which respectively measure the separation of proxy responses and their concentration. As shown in Fig. 6(a), existing methods exhibit high entropy with low contrast, indicating accumulated proxy activations and weak competition. In contrast, DDSPH achieves higher response contrast with moderate entropy, suggesting clearer proxy separation without collapsing to a single proxy. Fig. 6(b) further shows that DDSPH consistently shifts response contrast toward higher values across samples. These results indicate that DDSPH transforms proxy responses from accumulation to structured contrast, leading to clearer decision regions that complement the observed retrieval improvements.

### 4.7. Efficiency Analysis

We compare the training and encoding efficiency of DDSPH with DSPH, DNPH, and DHaPH under the same experimental setting in Fig. 7. DDSPH shows a clear advantage in training time, which is mainly attributed to its sample-specific proxy structuring that selectively aggregates a small subset of relevant proxies, avoiding exhaustive proxy–sample interactions. In contrast, existing proxy-based methods rely on global proxy alignment or neighborhood/hierarchy modeling, leading to higher computational overhead during optimization. For encoding, DDSPH achieves a modest speedup, as inference only involves a lightweight structured aggregation without introducing additional test-time complexity.

### 4.8. Visualization

To assess the discriminative structure of the learned hash codes, we apply t-SNE (Van der Maaten & Hinton, 2008) on MIRFLICKR-25K 32-bits and visualize the resulting embeddings for DDSPH and baseline methods in Figure 8. The

*Table 3.* mAP (%) under different loss weighting strategies at 32 bits. "Norm." denotes the proposed loss normalization in Eq. (10), which dynamically rescales each term by its own magnitude.

| Loss Weights | | | MIRFLICKR-25K | | MS COCO | |
|---|---|---|---|---|---|---|
| $\lambda_a$ | $\lambda_r$ | $\lambda_p$ | $I{\rightarrow}T$ | $T{\rightarrow}I$ | $I{\rightarrow}T$ | $T{\rightarrow}I$ |
| 1 | 1 | 1 | 85.52 | 83.68 | 74.12 | 74.18 |
| 0.1 | 1 | 1 | 85.81 | 83.92 | 74.38 | 74.42 |
| 1 | 0.1 | 1 | 85.75 | 84.00 | 74.29 | 74.35 |
| 1 | 1 | 0.1 | 85.73 | 83.84 | 74.25 | 74.29 |
| **Norm. (Ours)** | | | **85.99** | **84.05** | **74.59** | **74.66** |

*Table 4.* mAP results (%) of DDSPH and baselines under different noise rates on the MIRFLICKR-25K dataset w.r.t. four bits.

| Task | Method | 16 bits | | | 32 bits | | | 64 bits | | | 128 bits | | |
|------|--------|---------|---|---|---------|---|---|---------|---|---|----------|---|---|
| | | *0.2* | *0.5* | *0.8* | *0.2* | *0.5* | *0.8* | *0.2* | *0.5* | *0.8* | *0.2* | *0.5* | *0.8* |
| Image ↓ Text | NRCH | 76.38 | 75.61 | 73.24 | 76.75 | 76.21 | 74.76 | 77.05 | 76.98 | 76.07 | 77.83 | 77.10 | 76.11 |
| | DNPH | 77.70 | 75.55 | 75.37 | 79.74 | 80.16 | 79.04 | 81.67 | 82.44 | 81.01 | 83.53 | 83.32 | 81.08 |
| | DHaPH | 83.73 | 81.64 | 78.04 | 82.94 | 78.11 | 75.82 | 82.10 | 79.17 | 76.85 | 82.64 | 79.77 | 77.28 |
| | BiLGSEH | 78.11 | 77.35 | 75.93 | 78.24 | 77.36 | 75.15 | 80.72 | 78.42 | 76.30 | 80.00 | 79.32 | 77.46 |
| | DPBE | 77.85 | 77.16 | 73.79 | 80.80 | 77.16 | 74.09 | 82.21 | 78.28 | 75.45 | 84.12 | 80.11 | 76.77 |
| | DDBH | 83.79 | 82.20 | 76.50 | 84.70 | 82.11 | 78.78 | 85.04 | 84.58 | 79.46 | 85.64 | 84.61 | 80.29 |
| | **DDSPH** | **83.81** | **83.68** | **81.16** | **85.91** | **85.58** | **83.35** | **86.60** | **86.50** | **84.76** | **87.29** | **86.89** | **85.42** |
| Text ↓ Image | NRCH | 74.55 | 74.31 | 72.20 | 75.53 | 74.68 | 72.59 | 75.88 | 75.41 | 74.59 | 75.71 | 75.80 | 74.64 |
| | DNPH | 76.40 | 75.25 | 75.18 | 78.56 | 79.28 | 77.05 | 80.56 | 80.18 | 79.25 | 81.26 | 80.76 | 79.45 |
| | DHaPH | 81.51 | 79.49 | 77.66 | 80.52 | 75.96 | 73.22 | 78.69 | 76.39 | 74.23 | 79.31 | 76.99 | 74.38 |
| | BiLGSEH | 77.23 | 76.61 | 76.30 | 81.27 | 77.85 | 72.16 | 79.21 | 75.67 | 75.14 | 79.55 | 77.36 | 76.72 |
| | DPBE | 76.26 | 75.82 | 73.29 | 78.83 | 76.28 | 74.38 | 81.17 | 78.13 | 75.76 | 82.59 | 80.14 | 77.22 |
| | DDBH | 82.16 | 80.27 | 77.07 | 82.34 | 81.22 | 79.62 | 84.24 | 82.79 | 79.77 | 83.76 | 82.69 | 79.84 |
| | **DDSPH** | **82.61** | **82.69** | **80.93** | **84.30** | **84.40** | **82.50** | **84.99** | **85.20** | **83.46** | **85.59** | **85.70** | **84.26** |

The best and second-best performances are highlighted in **boldface** and underlined.

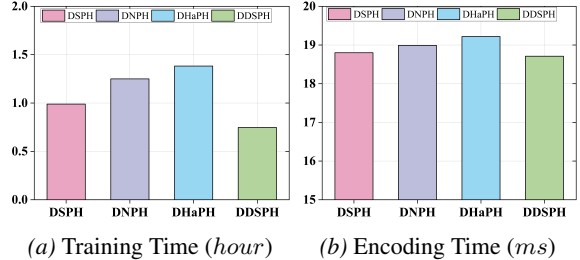

*(a)* Training Time (*hour*)    *(b)* Encoding Time (*ms*)

*Figure 7.* Training time of 100 epochs and encoding time of query set on MIRFLICKR-25K 32bits.

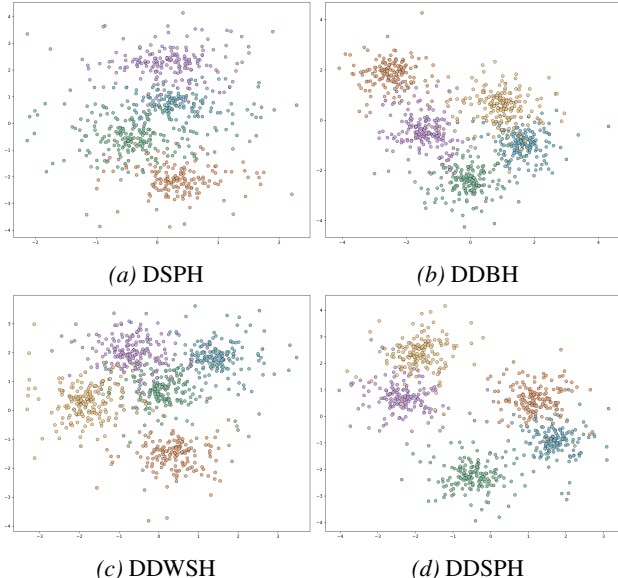

*(a)* DSPH    *(b)* DDBH

*(c)* DDWSH    *(d)* DDSPH

*Figure 8.* T-SNE results of DDSPH and comparative methods on MIRFLICKR-25K 32 bits.

embeddings learned by DDSPH exhibit more compact intra-class clusters and clearer inter-class separation than those of the baselines. Methods that treat proxies as independent semantic attractors produce embeddings where semantically distinct categories overlap in low-density boundary regions, reflecting the limited discriminative structure available when proxy responses are accumulated rather than contrasted. The structure-guided proxy competition in DDSPH explicitly contrasts proxy responses during optimization, producing a space in which category-level boundaries are more faithfully preserved across the full embedding distribution.

## 5. Conclusion

This paper presents DDSPH, a structure-discriminative proxy hashing framework for multi-label cross-modal retrieval, which organizes proxies into sample-specific relational structures to induce implicit cross-category competition and clearer decision boundaries in the Hamming space. Extensive experiments demonstrate consistent improvements across hash lengths while reducing reliance on a large number of proxies, indicating a favorable trade-off be-

tween efficiency and discriminative capability. Nevertheless, DDSPH has notable limitations. The subgraph construction relies on a selection guided by label masks, which may be suboptimal when label annotations are noisy or incomplete. Moreover, performance gains are less pronounced under high semantic complexity, as evidenced by the reduced margin on MS COCO at short hash codes, where the large category space constrains the discriminative capacity of compact representations. Future work will investigate adaptive subgraph construction and structured proxy interactions tailored to larger-scale and noisier supervision scenarios.

## Impact Statement

This paper presents work whose goal is to advance the field of machine learning. There are many potential societal consequences of our work, none of which we feel must be specifically highlighted here.

## Acknowledgments

This work was supported by the National Natural Science Foundation of China (No.62302338, No. 62472390), Shandong Provincial Natural Science Foundation (No.ZR2025MS1067).

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
