# OpenReview forum: "Deep Discriminative Structure Proxy Hashing for Cross-modal Retrieval"
_ICML.cc/2026/Conference — ICML 2026 regular_

### Official Review · Reviewer_YhKt · 2026-02-18

**Soundness:** 3
**Presentation:** 3
**Significance:** 2
**Originality:** 2
**Overall Recommendation:** 3
**Confidence:** 2

**Summary:**

The paper proposed DDSPH, a proxy-hashing method that organizes sample-specific proxies into structured and competitive relations. The proposed method maintains the structure of the proxies which conceptually leads to better strucuture of the representation space since it is more organized.

**Compliance With Llm Reviewing Policy:**

Affirmed.

**Final Justification:**

My concerns have been adequately addressed, and I tend not to change my score.

**Key Questions For Authors:**

1. The lines in Figure 1 are not understandable. What do red, blue, and purple lines mean?

See weakness 1

**Limitations:**

The proposed method has more components than previous works, which may make it run and train longer.
It will be good if the authors can share comparisons of the time/space complexity and inference time between the baselines and DDSPH.

**Strengths And Weaknesses:**

### Strengths:
1. The motivation is simple, and the used learning pipeline is too.
2. The paper approaches an NP-complete problem with a learnable framework with approximations.


### Weaknesses:
1. The performance of DDSPH vs. DDBH is not significantly better across datasets. Is there any way the authors can convince the reader that their method is better?
2. The motivation of the paper is simple, which has limited novelty in terms of scientific breakthrough.
3. The full name of DDSPH is never mentioned. Is it "Deep Discriminative Structure Proxy Hashing"?
4. Figure 3 is unreadable.

#### minor
1. Typo in line 316. set. set...

---

> ### Author Rebuttal · Authors · 2026-03-31
>
> ### W1 & Q2
>
> We conducted paired t-tests (10 random seeds) comparing DDSPH against DHaPH, DDBH, and DDWSH at 32 bits across all three datasets and both retrieval directions. Results are as follows. All differences are statistically significant at the 95% confidence level. We will incorporate standard deviations and significance annotations into Table 1 in the revision.
>
> |Methods||I2T|||T2I||
> |-|-|-|-|-|-|-|
> ||FLICKR|NUSWIDE|COCO|FLICKR|NUSWIDE|COCO|
> |vs.DHaPH|1.46E-06|0.0006|7.31E-06|8.63E-11|2.06E-06|5.18E-13|
> |vs.DDBH|0.0047|0.0002|0.0374|6.75E-05|0.0006|2.25E-09|
> |vs.DIMCH|1.81E-07|9.98E-10|1.39E-08|7.93E-07|3.16E-09|1.44E-09|
> |vs.DDWSH|6.23E-05|1.08E-05|0.0135|1.25E-05|0.0050|1.01E-05|
>
> Beyond mAP, DDSPH demonstrates a more substantial advantage in Hamming space quality: in Table 2, DDSPH consistently outperforms DDBH in the F@H metric across all three datasets, with the gap being most pronounced on MS COCO where semantic overlap is most complex. We believe this complementary evidence more directly reflects the method's contribution to discriminative representation learning.
>
> ### W2
>
> We respectfully note that conceptual simplicity and scientific contribution are not in conflict. The novelty of DDSPH lies in identifying a specific and previously unaddressed failure mode — proxy response accumulation — that is present across all existing proxy-based hashing methods regardless of their structural designs (neighborhood-aware, hierarchy-aware, or data-aware). To our knowledge, no prior work has approached proxy-based hashing from the perspective of how proxy responses are consumed rather than how proxies are constructed. The sample-specific subgraph mechanism is also introduced for the first time in this work.
>
> ### W3 & W4 & W5
>
> The full name appears in the paper title, but we acknowledge it was not expanded at first use. We will add the full name in the revision. We agree that Figure 3 in its current form is too dense to read at print size, and we apologize for this presentation issue. We will completely redraw the figure with substantially larger fonts, reduced visual clutter, and cleaner layout. Ensuring that all figures are legible at single-column width is a priority in the revision. We also confirm that "set. set..." is a duplication error. We have corrected in the revision and have fix any remaining typos and grammatical issues.
>
> ### Q1 & W6
>
> The three line colors in Figure 1 are used to convey that a sample can interact with multiple proxies simultaneously through the graph structure, capturing richer and more complex relational information than a single nearest-proxy assignment. Specifically, **red** lines indicate the most relevant proxy connections (highest similarity), **blue** lines indicate secondary relevant connections, and **purple** lines represent additional proxies that are incorporated into consideration by virtue of their position within the graph structure. The intent of using distinct colors is to illustrate that information propagates through a relational graph among proxies rather than through isolated point-to-point links — the emphasis is on the structural nature of the interaction as a whole, rather than on the specific differences between individual connection types, which is why we did not include a legend in the original figure.
>
> ### W7
>
> Fig.6 in the paper already reports training and encoding time on COCO 32bits:
>
> |Method|TrainTime(h)|EncodeTime(ms)|
> |-|-|-|
> |DSPH|0.989|18.80|
> |DNPH|1.250|18.99|
> |DHaPH|1.383|19.22|
> |DDBH|1.700|18.50|
> |DDSPH|0.748|18.71|
>
> DDSPH achieves the shortest training time among all compared methods, including simpler baselines such as DSPH, and its encoding time is on par with all baselines. Meanwhile, we also compared our chosen parameter configuration (M=3, r=0.1) with other high-performance parameter configurations.
>
> |M|r|I2T|T2I|TrainTime|
> |-|-|-|-|-|
> |3|0.05|84.56|83.22|1.18|
> |3|0.1|84.59|82.85|1.21|
> |7|0.1|84.72|82.89|1.84|
> |7|0.2|84.69|82.86|1.86|
>
> Furthermore, we provide a complexity breakdown for each training step:
>
> Space complexity: DDSPH introduces one additional parameter matrix — the proxy set P ∈ R^{K×CM}. Relative to DSPH which maintains P ∈ R^{K×C}, this is an M-fold increase in proxy parameters. With M = 3, this amounts to a modest and fixed overhead that does not scale with dataset size. All other network components (hash layers, backbone) are identical to the baselines.
>
> Time complexity per training step:
>
> |Step|Complexity|
> |-|-|
> |Similarity matrix|O(B×CM×K)|
> |graph construction|O(B×CM×log(CM))|
> |Label aggregation|O(B×k×C)|
> |Loss|O(B×C) for La;O((CM)²)for Lr;O(B²)for Lp|
>
> The only step not present in baseline methods is the top-k subgraph, which involves a partial sort over CM per sample, a negligible overhead in practice, consistent with the empirical training time being lower than all baselines. At inference, subgraph construction is not performed; hash codes are produced by a single forward pass through the hash layer.

---

> > ### Author Rebuttal · Reviewer_YhKt · 2026-03-31
> >
> > My concerns have been adequately addressed.

---

> > > ### Author Response · Authors · 2026-04-03
> > >
> > > We sincerely thank the reviewer for the careful re-evaluation and for confirming that the concerns have been adequately addressed. We are glad the rebuttal was helpful. Given that the reviewer notes all concerns are fully resolved, we would kindly ask whether the score might be reconsidered to reflect this, as it would more accurately represent the current state of the paper. We leave this entirely to the reviewer's discretion and appreciate the time invested in reviewing our work.

---

### Official Review · Reviewer_7S3p · 2026-03-03

**Soundness:** 2
**Presentation:** 2
**Significance:** 3
**Originality:** 2
**Overall Recommendation:** 4
**Confidence:** 3

**Summary:**

This paper proposes an innovation to proxy-based hashing for cross-modal retrieval of multi-label samples. The authors argue that in traditional proxy-based methods, each proxy acts independently to attract related samples and that since a given sample can possess multiple labels, it is pulled by multiple proxies without any clear competition, leading to poor discriminative structure for effective retrieval. To address this issue, the paper introduces a novel proxy-learning methodology named Deep Discriminative Structure Proxy Hashing (DDSPH). This method, during training, assigns a set of $k$ most relevant proxies (pre-filtered to the set of positive proxies) for a given sample. A three-pronged training objective is then applied over this filtered set of proxies and the corresponding samples in a given batch, to (1) align samples with proxies, (2) bring positive proxy pairs close together while pushing negative pairs apart, and (3) bring positive sample pairs close together while pushing negative ones apart. The learned embedding space is evaluated for cross-modal retrieval on three public cross-modal retrieval benchmarks to demonstrate the superiority (in terms of mAP@all) of the proposed DDSPH method over existing proxy-based methods. Ablations are performed toward analysing the structure of the learned Hamming space and the method's sensitivity to hyperparameters and supervision noise.

**Compliance With Llm Reviewing Policy:**

Affirmed.

**Final Justification:**

I appreciate the authors' sincere and thorough engagement with my review as well as the follow-up questions. As stated in my acknowledgement, many of my concerns have been addressed.

My remaining reservation (a soundness/significance concern) remains around the argument for the necessity of sample-specificity. I agree with the authors that  underperformance at $r=1$ and the higher contrast among proxy samples (the authors mention Figure 7 but probably meant Figure 6) hint towards the functional significance of sample-specificity; however, since this part is intended as a core contribution, I believe it remains quite necessary to better understand its influence on cross-modal retrieval. A direct study linking region dominance and proxy response strength and confirming the authors' intuition was probably not possible within the scope of a rebuttal, but the lack of it does undermine the contribution somewhat, in my assessment.

That said, the ablations added during the rebuttal have convinced me that the paper clears the bar for acceptance and I am happy to raise my overall recommendation to Weak Accept.

**Key Questions For Authors:**

My main questions / concerns / requests for clarification can be found under Weaknesses. Additionally,

1. Figure 2: Was "simple" meant to be "sample"?
1. Line 165, left: cosine similarities are already bounded within $[-1, 1]$; the mention of "normalization operations" (which are already implicit in the cosine computation) here seems redundant. If the authors meant something else, I would appreciate a clarification.
1. Table 1: If possible, it would be great to see the results after accounting for standard deviation across seeds used during training.
1. Line 374: how were $M$ and $r$ selected? Was the selection done post-hoc i.e., after performing the ablation or ad-hoc? If ad-hoc, why were these specific values selected?
1. Could the authors provide intuition (empirical or otherwise) as to which kinds of samples with the same multi-hot label are expected to select similar vs. different top-k proxy sets?

**Limitations:**

Limitations were not explicitly discussed, to the best of my knowledge.

I also did not find an Impact Statement (as required by the ICML author guidelines) in the paper.

**Strengths And Weaknesses:**

### Strengths

**Significance**: Efficient cross-modal retrieval is an important area of application; the work positions itself as an improvement over existing methods in this body of research.

**Soundness**: Both the problem (fast cross-modal retrieval) and the high-level solution (incorporating structure into proxies) make conceptual sense. In terms of mAP, the method largely outperforms existing works, indicating that the proposed sample-specific proxy learning method has merit (although with caveats I mention later). The ablations are fairly extensive: for instance, they reasonably establish the importance of each of the three loss terms.

**Presentation**: The paper is well-written and provides sufficient background.

### Weaknesses

**Soundness**:

1. While DDSPH achieves better retrieval results, the claim that the method discovers "sample-specific proxy relations" seems overstated. The "competition" among proxies comes primarily from the initial top-k filtering among positives (negatives are preemptively eliminated from consideration); if there is another source of competition beyond the filtering, I missed it and would appreciate a clarification from the authors. In fact, the work places significant emphasis on existing works adding similarities with proxies rather than contrasting them; however, after the top-k filtering, the current method still resorts to addition (e.g., during label aggregation).

1. In practice, it is also not clear why different samples with the same multi-hot label would have different top-k proxies and whether the top-k filtering actually contributes to the performance gain. While the motivation and novelty of the paper rely a lot on this filtering step, neither of these aspects is explored in quantitative/qualitative ablations. For instance, it would be worth exploring whether a lower $M$ with $r = 1$ achieves performance competitive with the case where top-k filtering is applied.

1. The loss normalization scheme in Eqn. 10 is, to the best of my knowledge, not ablated against simpler schemes like uniform weighting; hence, the current choice seems arbitrary.

1. It is unclear from the paper whether the baselines use the same backbone (CLIP) as the proposed method: this information is crucial for contextualizing the role of the underlying pretrained embeddings towards the observed performance gains. The specific CLIP model used in the experiments was not mentioned; it is also unclear whether the conclusions would hold for other CLIP models or other classes of vision-language models altogether. Depending on whether all baselines use the same underlying model, this might be more of a clarity concern than a soundness concern; nevertheless, it remains crucial to my assessment of the paper.

1. The main results (Table 1) are also missing statistical significance testing; this is quite important given that DDSPH often outperforms baselines by small differences in mAP.

**Presentation**: Figures would greatly benefit from improvement (e.g., font sizes in Figures 3, 5, etc.). The paper can certainly benefit from typo fixes and clearer writing (e.g., referring to "relations" as singular instead of plural; broken sentence structure in line 139, right; repetition in line 178, left; "comparison" --> "compare" in line 391, right; etc.).

Overall, while the proposed method shows clear empirical gains in retrieval performance, I am not yet convinced that its (implementational / conceptual) complexity is sufficiently justified by the ablations, or that some of the framing (e.g., sample-specific proxy relations) is accurate. At this point, my assessment of the paper is likely to change only if most of the concerns raised here are addressed.

---

> ### Author Rebuttal · Authors · 2026-03-31
>
> ### W1&Q5
> The structural competition in DDSPH is not reducible to top-k filtering alone. In conventional multi-center proxy methods, proxies operate independently per category, and cross-category responses are accumulated without any structural coupling. DDSPH differs in that the top-k selected proxies are drawn from multiple co-activated categories and organized into a shared subgraph. Within this subgraph, all selected proxies jointly participate in label aggregation, meaning the relative contribution of one category's proxy is structurally coupled to others; a higher response from one proxy reduces the relative weight of others, and gradients flow back jointly. This cross-category coupling constitutes implicit competition for relative influence within a capacity-constrained shared structure, which is absent in all prior proxy-based methods. Negative supervision is handled outside the subgraph via Lr and Lp.
>
> DDSPH's sample-specificity operates at the cross-category level rather than the intra-class level. Under multi-label co-activation, the relative dominance among co-activated categories varies across samples, even with identical multi-hot labels: a "dog+lake" sample dominated by the dog region assigns higher similarity to dog-category proxies, reshaping the subgraph composition differently from a lake-dominated sample with the same label. In conventional methods, this difference only affects which proxy within each class is selected, while cross-category responses remain independently accumulated. In DDSPH, it reshapes the entire subgraph, the relative weights among cross-category proxies and the joint gradient signal both differ, producing a sample-specific decision region that independent per-class assignment cannot achieve.
>
> ### W2
>
> We confirm that r=1 was validated during method development and showed a clear performance drop. When organizing the parameter analysis, we inadvertently omitted it as a boundary baseline; the updated results, including r=1 are provided in RE:W1 of Reviewer ZGRj. M controls the total representativeness of the proxy set; reducing its value limits intra-class coverage, which is costly in multi-label settings with high within-class diversity. The ratio r controls structural sparsity at the per-sample interaction level without affecting proxy capacity. A smaller r with sufficient M preserves broad semantic coverage while enforcing focused interactions per sample; collapsing both into a smaller M with r=1 sacrifices coverage and selectivity simultaneously.
>
> ### W3
>
> The results on FLICKR 32-bit are as follows. The final config("-") normalizes each loss term by dividing by its own scalar magnitude at each training step, effectively bringing all three terms to a comparable unit scale dynamically rather than assigning fixed coefficients. This avoids the need to manually balance loss magnitudes and proves more effective than any of the fixed-weight alternatives above. The three losses are naturally suited to this treatment since they all operate on normalized similarity scores, keeping their magnitudes in a compatible range throughout training. We will include this analysis.
>
> |λ₁|λ₂|λ₃|I2T|T2I|
> |-|-|-|-|-|
> |1|1|1|85.52|83.68|
> |1.0|0.1|1.0|85.75|84.00|
> |1.0|1.0|0.1|85.73|83.84|
> |0.1|1.0|1.0|85.81|83.92|
> |-|-|-|85.99|84.05|
>
> ### W4
>
> All baseline methods and DDSPH use CLIP ViT-B/32 as the backbone with official pretrained, consistent with the recent works. We will further state this in the revision. To address the reviewer's concern about generalization to other vision-language models, we conducted additional experiments on COCO across CLIP Res50, BLIP I-C-Base, and SigLIP2 Base-32. Results shows mAP is relatively stable across models.
>
> ### W5&Q3
>
> We conducted paired t-tests in RE:W1 of Reviewer YhKt. DDSPH achieves statistically significant improvements over most baselines on the majority of datasets and tasks.
>
> ### Presentation&Q1&Limitation
>
> We appreciate the feedback and have rechecked and corrected the typos, added the Impact Statement.
>
> ### Q2
>
> Our intention was to emphasize the normalization step explicitly, but given that conclusion can be easily drawn and has no further significance, this supplementary note appears to create confusion rather than aid clarity. We will remove this phrase in the revision.
>
> ### Q4
>
> The parameters M=3 & r=0.1 were selected post-hoc, based on a joint consideration of retrieval performance and training efficiency. While certain configurations achieve marginally higher mAP, they also incur substantially higher computational cost. The selection represents the configuration that achieves a favorable balance between the two. This efficiency-aware selection rationale was part of our decision process but was omitted from the paper due to space constraints. We provide the key supporting data in RE:W7 of Reviewer YhKt. M=3,r=0.1 achieves performance at roughly two-thirds the training cost, making it the practical optimum. We will include this analysis.

---

> > ### Author Rebuttal · Reviewer_7S3p · 2026-04-02
> >
> > Thank you for the rebuttal. The additional experiments addressed many of my concerns.
> >
> > However, regarding W1, I would appreciate some additional clarifications.
> >
> > 1. I understand that the contributions from the proxies are learned, and they compete against each other towards the aggregation. But I'd appreciate the authors clarifying the differences between their proposed method and Qian et al. (Deep metric learning without triplet sampling, ICCV 2019) who also apply softmax over multiple "centers" per class (as noted in §2.2 of the submission). Do I understand correctly that the novelty lies in that DDSPH applies this idea to the setting where multiple classes can activate for the same sample?
> >
> > 2. The motivation for subgraph construction (§3.3) and its downstream use is unclear to me. Could the authors clarify the role of a subgraph, especially a *directed* one (line 156), and what the directions in the edges encode?
> >
> > 3.
> >
> > > Under multi-label co-activation, the relative dominance among co-activated categories varies across samples, even with identical multi-hot labels: a "dog+lake" sample dominated by the dog region assigns higher similarity to dog-category proxies, reshaping the subgraph composition differently from a lake-dominated sample with the same label.
> >
> > This sounds reasonable and helps my understanding of the sample-specificity reasoning. However, were there empirical studies (qualitative or otherwise) to verify this intuition? Perhaps, linking the fraction of the image taken up by a particular category (using segmentation masks for instance) to the order of proxy response strengths? I believe this would be quite helpful towards motivating the setup.

---

> > > ### Author Response · Authors · 2026-04-03
> > >
> > > We thank the reviewer for the continued engagement and the thoughtful follow-up questions.
> > >
> > > ---
> > >
> > > **Q1: Relationship to Qian et al. (SoftTriple, ICCV 2019)**
> > >
> > > The reviewer's understanding is correct, and we appreciate the precise formulation. SoftTriple applies softmax normalization over multiple centers *within* a single class, so competition is strictly intra-class: centers from different classes never interact within the same normalization scope, and cross-category responses are still accumulated independently. DDSPH extends this idea in a fundamentally different direction: rather than normalizing within each class separately, we organize proxies from *multiple co-activated classes* into a shared subgraph for each sample, and normalization (degree normalization in Eq. 5) operates over this cross-category structure jointly. The competition therefore occurs across class boundaries rather than within them, which is the setting that multi-label co-activation uniquely demands. We will add a direct comparison to SoftTriple in the related work section to make this distinction explicit.
> > >
> > > ---
> > >
> > > **Q2: Role and directionality of the subgraph**
> > >
> > > The directed graph is constructed with edges pointing from each sample node toward its selected proxy nodes, encoding the direction of similarity-based influence: each edge weight represents how strongly a sample is drawn toward a given proxy, not a relationship between proxies themselves. The subgraph serves to define, for each sample, a sparse local neighborhood of semantically relevant proxies drawn from its co-activated categories. This neighborhood is then used for label aggregation (Eq. 5), where the edge weights (similarity scores) determine each proxy's contribution to the sample's prediction. The directionality simply reflects that this is a sample-to-proxy relationship rather than a symmetric or proxy-to-proxy one. We will clarify this in the revision in Section 3.3 by explicitly describing the edge semantics.
> > >
> > > ---
> > >
> > >  **Q3: Empirical verification of sample-specificity intuition**
> > >
> > > We appreciate the reviewer's push for more direct evidence. We want to clarify that sample-specificity is an emergent property of the similarity-based subgraph construction rather than an explicitly designed objective — samples with the same multi-hot label naturally produce different subgraphs because their feature representations differ. The more precise empirical question is therefore not whether this property exists, but whether the subgraph structure meaningfully exploits it to produce better representations. Two pieces of existing evidence bear on this. First, the parameter study shows that r = 1.0 (where all positive proxies enter the subgraph regardless of sample-specific similarity ranking, minimizing the effect of sample-specificity) consistently underperforms r < 1 across all values of M, confirming that selective, sample-specific subgraph construction contributes positively beyond simply having more proxies. Second, Fig. 7 shows that DDSPH produces substantially higher proxy-sample response contrast compared to baselines, indicating that the subgraph structure produces differentiated rather than uniform proxy activations — which is precisely what sample-specific subgraph composition would induce. We agree that a direct study linking region dominance to proxy response strength (e.g., via segmentation masks) would be an illuminating future analysis, and we will note this as a direction in the revision. However, we believe the above evidence sufficiently supports the functional role of sample-specificity within the proposed framework.
> > >
> > > ---
> > >
> > > We sincerely thank the reviewer for the continued engagement and the constructive  follow-up questions, which have helped us sharpen both the claims and the presentation  of the paper. We hope the above clarifications are satisfactory, and we would kindly  ask the reviewer to consider whether the responses warrant a revision of the score.

---

### Official Review · Reviewer_ZGRj · 2026-03-13

**Soundness:** 3
**Presentation:** 3
**Significance:** 3
**Originality:** 3
**Overall Recommendation:** 4
**Confidence:** 4

**Summary:**

This paper proposes DDSPH, a proxy-based cross-modal hashing method that argues existing proxy methods mainly accumulate responses from multiple proxies, which may hurt discriminability in multi-label retrieval. To address this, the paper introduces sample-specific proxy structures so that relevant proxies are explicitly contrasted rather than only aggregated, and experiments on three benchmarks with multiple metrics suggest the method is effective.

**Compliance With Llm Reviewing Policy:**

Affirmed.

**Final Justification:**

I keep my original score.

**Key Questions For Authors:**

Questions to the Authors:
1. The paper’s core idea is that proxy responses should be contrasted rather than accumulated, and this is tied to the claim of better discriminability. Could the authors provide a more intuitive visualization, such as a scatter plot of the learned representations or Hamming-space distributions, to make this claim more directly visible? Right now the paper argues for improved discriminability, but a more direct visual demonstration would make the claim much more convincing.
2. I found the current figure design somewhat problematic. In Fig. 5(b), the axis scale differences are very small, which visually blurs the gap between methods; this feels less informative than a simple table would be. Similarly, Fig. 4 could likely be replaced with a standard bar chart that uses much less space while being more straightforward. Also, the axis label “M” should probably be changed to “m” for consistency with Eq. (9). This is a conference paper rather than a presentation slide deck, so direct readability should take priority over visual styling.

Additional Discussion (no need to respond):
One broader point I kept thinking about is whether discriminability should be treated as the central objective as strongly as the paper suggests. Retrieval is not the same as classification: in retrieval, especially under multi-label settings, semantic overlap is inherent, and the retrieval database itself may not admit sharply separable boundaries in the same way as a classification problem. From that perspective, it is possible that what matters most is not necessarily making the space more “discriminative” in a strict sense, but learning a better structured distribution that preserves graded semantic relatedness [1]. I think the paper would benefit from discussing this distinction more explicitly, since it would also help clarify what exactly DDSPH is improving.

[1] Cross-Modal Hashing Method with Properties of Hamming Space: A New Perspective, TPAMI, 2024.

**Strengths And Weaknesses:**

Strengths:
1. The paper has a clear motivation, and the method is designed in a way that is well aligned with that motivation.
2. The evaluation is relatively complete, going beyond mAP and including additional metrics and analyses that try to validate the paper’s discriminability claim from multiple angles.

Weaknesses:
1. Although some figures appear carefully designed, they sometimes hurt rather than help intuitive understanding. In particular, Fig. 4 and Fig. 5 are less direct than they could be, and the current visual presentation makes it harder to quickly extract the main information.
2. Some claims in the paper are stronger than the evidence fully supports, especially the broader interpretation that the gains mainly come from improved discriminative structure rather than other possible architectural or optimization effects.
3. The presentation can be improved. There are a number of awkward phrasings and writing issues in the method section, which occasionally make the technical details harder to follow than necessary. Also, some punctuation around Eqs. (1)–(6) is missing, which further affects readability.

---

> ### Author Rebuttal · Authors · 2026-03-31
>
> We thank the reviewer for the positive assessment and the thoughtful suggestions. We respond to each point below.
>
> ---
>
> ### W1 & Q2: Figure Design & M/m
>
> We appreciate the careful reading, but we want to clarify that **m** and **M** refer to two distinct quantities and the difference is intentional. In Eq. (9), **m** is the margin hyperparameter in the triplet loss, which is set to the standard value of 0.25 and was not discussed separately as it follows common practice. **M** in Fig.5 refers to the number of proxies per category, as defined in Section 3.1 of the method. We have replaced “M” with “N” to make the wording more natural and avoid confusion. In the rebuttal, we will continue to use M for the time being.
>
> Regarding figure design: We agree that Figure 4 would be clearer as a standard bar chart and will replace it accordingly. For Fig.5, we agree that the axis scale differences are subtle and will replace the plot with a numerical table that includes standard deviations, which is more informative in a paper format.
>
> Here is the original table from Fig.4-5:
>
> Ablation study — I→T mAP on MIRFLICKR-25K
> |Bits|DDSPH|w/o La|w/o Lr|w/o Lp|
> |-|-|-|-|-|
> |16|84.86|82.63|84.76|84.56|
> |32|85.99|84.11|85.32|85.65|
> |64|86.83|85.25|86.15|86.53|
> |128|87.53|86.20|86.73|87.16|
>
> Ablation study — T→I mAP on MIRFLICKR-25K
> |Bits|DDSPH|w/o La|w/o Lr|w/o Lp|
> |-|-|-|-|-|
> |16|83.22|82.00|82.95|82.85|
> |32|84.05|83.02|83.24|83.19|
> |64|84.90|83.29|83.96|83.69|
> |128|85.52|84.20|84.42|84.23|
>
> Parameter study — I→T mAP on MIRFLICKR-25K (32-bit), varying M and r
> |r\M|M=1|M=2|M=3|M=4|M=5|M=6|M=7|M=8|
> |-|-|-|-|-|-|-|-|-|
> |r=0.05|84.33|84.29|84.56|84.14|84.39|84.25|84.14|84.42|
> |r=0.1|84.67|84.66|84.59|84.21|84.24|84.15|84.72|84.19|
> |r=0.2|84.52|84.50|84.62|84.20|84.31|84.19|84.69|84.36|
> |r=0.5|84.39|84.41|84.58|84.12|84.29|84.26|84.50|84.32|
> |r=1.0|83.87|83.94|84.11|83.76|83.89|83.81|84.03|83.92|
>
> Parameter study — T→I mAP on MIRFLICKR-25K (32-bit), varying M and r
> |r\M|M=1|M=2|M=3|M=4|M=5|M=6|M=7|M=8|
> |--------|-----|-----|-----|-----|-----|-----|-----|-----|
> |r=0.05|82.97|82.79|83.22|82.87|82.90|83.04|82.83|82.85|
> |r=0.1|83.14|83.03|82.85|83.03|82.79|82.99|82.89|82.66|
> |r=0.2|83.09|82.99|83.01|82.96|82.87|83.02|82.86|82.72|
> |r=0.5|83.02|82.89|83.16|82.89|82.91|82.99|82.90|82.80|
> |r=1.0|82.43|82.51|82.68|82.39|82.47|82.43|82.55|82.41|
>
> ---
>
> ### W2: Stronger Claims
>
> We appreciate this measured critique. We agree that the current phrasing overstates the causal attribution of performance gains to discriminative structure alone, without ruling out contributions from the loss design or subgraph regularization effect. We will revise the relevant claims to use more cautious language, such as "the results are consistent with the interpretation that..." and will acknowledge in the discussion that the observed improvements likely arise from the interplay of multiple design choices.
>
> ---
>
> ### W3: Presentation & Equations
>
> Thanks for the critique. We will revise the method section for clarity, particularly Section III. All missing punctuation around Eq. 1–6 will be corrected in line with standard mathematical writing conventions.
>
> ---
>
> ### Q1: Visualization
>
> Due to the space limitation of the main text, we did not include the visualization section in the manuscript. We have completed the t-SNE experiment comparing with the SOTA methods on the MS COCO 32-bit to show cluster separation. The results demonstrate that our method achieves good discriminative performance. We have also considered the Hamming distance distribution plot. However, since the F@H metric similarly calculates the Fisher ratio of distances between positive and negative samples in the embedding space, there is effectively functional overlap. But we still made it in the appendix of our revision.
>
> ---
>
> ### Additional discussion: Discriminability in Retrieval
>
> We thank the reviewer for raising this important distinction. We agree that retrieval under multi-label settings fundamentally requires preserving graded semantic relatedness rather than enforcing hard classification boundaries. DDSPH does not aim to produce a sharply partitioned space; rather, the structured proxy competition is designed to reduce undifferentiated response accumulation, which we argue degrades the quality of the learned distribution without improving semantic grading. The F@H metric we use measures the relative separation of positive and negative pairs, not a hard margin — improving F@H is compatible with preserving semantic gradients. We will explicitly discuss this distinction in the revision and cite the referenced work [1] to position our contribution more carefully.
>
> [1] Cross-Modal Hashing Method with Properties of Hamming Space: A New Perspective, TPAMI, 2024.

---

> > ### Author Rebuttal · Reviewer_ZGRj · 2026-04-05
> >
> > My issue has been resolved, and I stand by my original score.

---

### Decision · Program_Chairs · 2026-04-30

**Decision:**

Accept (regular)

**Comment:**

Following the rebuttal phase, the reviewer scores are 3, 4, and 4. After carefully examining the manuscript, the reviewers' comments, and the authors' response, I believe the paper's core motivation and the proposed approach require substantial visual evidence to be fully validated. Furthermore, I find the concerns raised by Reviewer 7S3p to be both reasonable and precise, and the authors' rebuttal does not completely resolve these issues. Weighing all the feedback, I lean towards acceptance. The paper only reach the threshold for Borderline Accept. I expect that the authors will incorporate the suggested visualizations and carefully address the remaining points in the camera-ready version.